# Heterologous RNA Recombination in the Cystoviruses φ6 and φ8: A Mechanism of Viral Variation and Genome Repair

**DOI:** 10.3390/v14112589

**Published:** 2022-11-21

**Authors:** Paul Gottlieb, Aleksandra Alimova

**Affiliations:** School of Medicine, The City College of New York, The City University of New York (CUNY), New York, NY 10030, USA

**Keywords:** heterologous recombination, homologous recombination, cystovirus, φ6, φ8, dsRNA

## Abstract

Recombination and mutation of viral genomes represent major mechanisms for viral evolution and, in many cases, moderate pathogenicity. Segmented genome viruses frequently undergo reassortment of the genome via multiple infection of host organisms, with influenza and reoviruses being well-known examples. Specifically, major genomic shifts mediated by reassortment are responsible for radical changes in the influenza antigenic determinants that can result in pandemics requiring rapid preventative responses by vaccine modifications. In contrast, smaller mutational changes brought about by the error-prone viral RNA polymerases that, for the most part, lack a replication base mispairing editing function produce small mutational changes in the RNA genome during replication. Referring again to the influenza example, the accumulated mutations—known as drift—require yearly vaccine updating and rapid worldwide distribution of each new formulation. Coronaviruses with a large positive-sense RNA genome have long been known to undergo intramolecular recombination likely mediated by copy choice of the RNA template by the viral RNA polymerase in addition to the polymerase-based mutations. The current SARS-CoV-2 origin debate underscores the importance of understanding the plasticity of viral genomes, particularly the mechanisms responsible for intramolecular recombination. This review describes the use of the cystovirus bacteriophage as an experimental model for recombination studies in a controlled manner, resulting in the development of a model for intramolecular RNA genome alterations. The review relates the sequence of experimental studies from the laboratory of Leonard Mindich, PhD at the Public Health Research Institute—then in New York City—and covers a period of approximately 12 years. Hence, this is a historical scientific review of research that has the greatest relevance to current studies of emerging RNA virus pathogens.

## 1. Introduction

Homologous and heterologous recombination in positive-strand RNA viruses, particularly poliovirus and coronavirus, has been identified and studied [1,2,3,4,5,6,7,8,9,10], and the reassortment of RNA segmented viruses is well documented [11,12]. RNA recombination in viral pathogens has been demonstrated to have huge medical, social, and political consequences throughout the world, as dramatically demonstrated by the current coronavirus pandemic [13]. Clearly, the study of RNA virus recombination is a subject of crucial importance that needs to be further analyzed. It is to this end that this review is directed. In a series of classic papers, Leonard Mindich of the Public Health Research Institute of New York (PHRI) described heterologous recombination in the segmented double-stranded RNA cystovirus bacteriophages φ6 and φ8. These seminal studies were performed, and the results were published beginning in 1992 and continued until 2004. Although much related research was performed in the Mindich laboratory regarding RNA packaging, transcriptional control, virus assembly, cystovirus protein structure, and the isolation of additional cystovirus species [14], this review focuses on the RNA intramolecular recombination (as opposed to segment reassortment) and how the structure of the procapsid (PC) facilitates this mechanism. Although reviews of the phenomenon were previously published by the principal investigator [14], a new summary of the discovery is now in order. Our review summarizes the major observations and conclusions with the aim of illuminating the ideas and concepts presented in this series of historical papers. The review can be used as a helpful guide to those inclined to read the papers in detail, in addition to providing an overview of a specific and insightful subject of study from the Mindich laboratory.

In light of the emergence of the COVID-19 pandemic caused by the human coronavirus pathogen SARS-CoV-2, the lesson provided by the cystovirus-based discoveries must be revisited. Although hardly evident 20 years ago, the subject of RNA genome reorganization has taken center stage, as the origin of the SARS-CoV-2 virus is a recognized result of genome recombination among mammalian coronaviruses [15]. It must be of great surprise to the virology community that the subject has even taken on geopolitical implications as debate on the SARS-CoV-2 origins in regard to both the recombinant mechanism and the location rages to the present moment [16,17]. The review takes no political position or assignment of blame to any laboratory, researcher, or nation but only describes the mechanisms guiding the exchange of genetic information as seen in φ6 and φ8 and speculates as to the potential relation of what is observed in coronaviruses. However, the structure and replicative cycle of the cystoviruses must first be understood.

## 2. The Cystovirus Structure and Replicative Cycle

The cystoviruses (family *Cystoviridae*) (species φ6–φ14, φYY, φNN, φ2954) are a unique group of viruses that have proven of great utility in the study of many facets of molecular virology [18,19]. Their replicative mechanism and structure make the species analogous to members of the *Reoviridae* family [20]; therefore, studies of the packaging and assembly of cystoviruses have shed light on analogous systems in the pathological reoviruses. Interestingly, the activity of the RNA-directed RNA polymerase (RdRP) has also been shown to have great similarity to the polymerase of flaviviruses [21]. Specifically, the cystoviruses are closest in form to the Group A rotaviruses that are the leading cause of severe dehydrating diarrhea in children under 5 years old in areas of the world where rapid medical intervention is difficult [22]. The cystovirus wildtype species all possess three double-stranded RNA (dsRNA) segments as their genome (known as the s, m, and l for the small, medium, and large segment transcripts and designated S, M, and L for the dsRNA genome segments. This nomenclature convention is used in the current review [18]. The architecture is shown schematically in Figure 1A. As described in the review, the PC can package an altered number of recombinant RNA genome segments under specific circumstances.

*Cystoviruses have a layered structure, and all species share similar structural features*. The only member of *Cystoviridae* family, the *Cystovirus*, is an enveloped bacteriophage of multilayered structure. The first species isolated, φ6, is the best characterized with regard to assembly and structure. The inner layer, the polymerase complex (PC), is composed of 120 units or 60 asymmetric dimers of protein P1 forming a dodecahedral T = 1 shell [23,24,25]. At each of the 12 dodecahedron faces, protein P4 hexamers (the packaging NTPase) protrude from the PC. The φ6 nucleocapsid (NC) shell is formed of 200 P8 trimers arranged in a T = 13 organization that partially covers the PC [23]. The outermost layer is a phospholipid bilayer membrane derived from the host cell cytoplasmic membrane. The host receptor spike protein, P3, extends from the viral outer envelope where it is anchored to the integral highly hydrophobic membrane protein P6.

*The layered structures are not identical among the species:* Although similar in overall design, the individual species of the cystoviruses are not entirely isomorphic. NC structures from φ8 and φ12 have recently been derived, and, while the overall organization of the dimers into a T = 1 icosahedral lattice seems similar in the three viruses (φ6, φ8, and φ12), the details of protein organization are clearly different. In φ8, the NC cover is virtually nonexistent with only 60 copies of P8 as a membrane-associated protein [26]. Our single-particle data place φ12 in an intermediate category with an incomplete T = 13 organization of the φ8 shell. This analysis demonstrated that the φ12 nucleocapsid protein P8 shell is composed of an incomplete T = 13 icosahedral lattice where the symmetry axes of the T = 13 and T = 1 layers superpose [26]. In addition, the outer surface binding structure differs radically from that of φ6 carrying horizontal spikes and a torroid complex structure on φ8 and φ12 [27]. This outer surface heterogeneity likely reflects the unique attachment sites of the viruses. φ6 attaches to the host cell-type IV pilus to initiate infection, while both φ8 and φ12 bind to a truncated lipopolysaccharide O chain (lps).

In a sense, the virus species of this family constitute a readymade mutant library in that they have a very similar overall genetic organization and express similar proteins from the three polycistronic mRNA that are (as mentioned above) indistinguishable from the packaged ssRNA. However, it must be emphasized that the primary genetic sequences of each of the three dsRNA segments differ to allow encoding of the early and late proteins. The structural proteins P1, P2, P4, and P7 that assemble the capsid are the early elements encoded on the L segment. The M segment carries the open reading frames for the envelope proteins P3, P6, and P10 with a late function, and the S segment encodes the outer protein matrix layer P8, a nonstructural protein thought to facilitate envelope assembly of P12 and the murein peptidase P5 utilized during viral entrance and exit from the host cell [28,29,30]. The organization of the entire viral genome is shown in Figure 1C.

The initially discovered and characterized cystoviruses infected strains of the plant pathogen, *Pseudomonas syringae pv. Phaseolicola* [18]. The host range of cystoviruses is governed by the structure of the attachment apparatus, as well as the specific cell surface receptor it utilizes. In the case of φ6 and its closest relatives, host infection requires binding to type IV pili [31,32]. Therefore, the phi6 group has a host range limited to *Pseudomonas syringae* and several selected mutants of Pseudomonas *pseudoalcaligenes* ERA [33]. φ8 and φ12 are considered distantly related to φ6, and they are able to infect *P. syringae* mutants by binding to rough lipopolysaccharide (LPS) [18]. Mutants lacking rough LPS, such as HB10Y, are not infected by φ8 and φ12. φ12 has the ability to infect other Gram-negative bacteria containing rough LPS, such as *E. coli* JM109, but without lytic plaque formation. In 2010, the *Cystoviridae* family was again expanded with the isolation of φ2954 from infected radish leaves [19]. The base sequences for many of the genes, and the segment termini were similar but not identical to those of bacteriophage φ12. However, the host specificity was for the type IV pili of *Pseudomonas syringae* HB10Y rather than the rough LPS to which φ12 attaches. φNN was later isolated from a freshwater habitat, demonstrating that the virus group is quite diverse and widely distributed [34]. This phage utilizes the type IV pilus of *Pseudomonas sp.* B314 for attachment, and phylogenetic trees of the genome segments indicate a close relationship to φ6. Notably the Cystovirus φYY, which uses the opportunistic pathogen *Pseudomonas aeruginosa* strain PAO38 as a host, was successfully isolated from hospital sewage in China in 2016 [35].

As a result of the genetic sequence analysis, cystovirus bacteriophages are divided into two major groups in which φ7, φ9, φ10, φ11, and φ2954 are closely related to φ6, while φ8, φ12, and φ13 are distantly related to φ6 [19,36,37,38]. The cystoviruses are all easily cultured and isolated, with only three RNA segments, and they are amenable to molecular dissection in vitro. In seminal studies, performed at the PHRI of NY, the structure, replication, genome packaging, and transcriptional regulation of φ6 were analyzed [39]. This research established the first in vitro RNA packaging and replication assay for this class of viruses and subsequently utilized it to construct infectious particles. In this seminal work, it was shown that cystoviruses package their RNA into preformed PCs, and it was demonstrated that φ6 packaging relies upon signals at the 5′ end (called pac) containing about 200 nucleotides with extensive secondary structure, while specific sequences at the 3′ end are RNA replication signals [40].

The packaging of the viral PC is sequential in the order S, M, and L, and capsid expansion accommodates the internalized RNA (Figure 1B) [25]. Once packaged and replicated to the mature double-stranded RNA genome, the tightly packed conformation is seen to have different liquid crystalline geometries and nonsymmetrical characteristics with pseudo-D3 symmetry forming five concentric layers [41,42]. This quality could have significance in the recombination mechanisms and will be discussed at the end of the review. The dynamics of the packaging and the protein components responsible were recently reviewed in detail, and the reader is referred to this publication for additional information [43]. However, now that the above brief review of the Cystoviridae has been presented, the discovery and description of cystovirus heterologous recombination can be better understood.

## 3. Reverse Genetics of φ6 and the Establishment of Carrier States

The establishment of an in vitro RNA packaging and viral rescue system allowed reverse genetics to be performed with φ6. The first reverse genetic system for a dsRNA segmented genome virus was established for the φ6 bacteriophage in 1990 (at this time, the only cystovirus identified was φ6) [44]. The rescue procedure (although technically cumbersome as compared to the later cystovirus rescue methods) required (a) the in vitro packaging of the three ssRNA transcripts into the empty recombinant PC, (b) coating the filled PC with a P8 protein lattice (assembling a NC in vitro), and (c) transfection into *P. syringae pv. phaseolicola* HBMPO.16 spheroplasts which were then plated on a lawn of intact *P. syringae pv. phaseolicola* HB10Y bacterial cells. Selected plaques were confirmed as the rescue noted by the small deletions inserted into the viral genome. Despite a modest beginning, the procedure shortly demonstrated its utility for the incorporation of foreign genes into the viral genome. Onodera et al. (1992) exploited the technique by inserting a transcript containing the kanamycin resistance gene (*kan*) into a viable virus and established a carrier state (CS) in the host cells [45]. While carrier states were previously described for φ6 and other lytic viruses, including reovirus [46,47,48], this study produced a unique transducing virus carrying the extra genetic material in the genome. In a sense, the virus acted as an RNA-based episome in that, although no assembled viral particles were produced, *kan* selection applied to the host cells. The fundamental observation was the demonstration of a CS in which the host pseudomonads were infected, and the viral genome replication was compatible with cell viability. However, at the time the study was performed, it remained unclear whether *all cells* produced infectious virus or even if the genome replicated in the CS remained entirely intact. The *kan* cartridge was seen to be genetically unstable, and the instability was postulated to result from either the burden of an extra 1.2 kbp contained in the M segment genome or the secondary structure of the excess RNA. Specifically, the *kan* gene cartridge, derived from plasmid pUC-4k (Amerham Biosciences), was bound by 12 G residues on the 5′ side and 12 C residues and a *Pst1* restriction site on the 3′ side. The sequence had the capability to form a hairpin structure that was suspected of being the cause of genetic instability. The instability was noted from the heterogenous plaque types resulting from propagation of the initial transductant (designated φ6 K1). Clear plaques were selected that yielded only additional clear plaques lacking the *kan* gene but smaller turbid plaque formers always yielded small turbid plaque progeny. RNA extracted from the turbid variety had normal-sized L and S segments, but the M segment was larger by 1.2 kbp accounting for the *kan* insert. Surprisingly clear plaques yielded RNA segments with normal-sized L and S, but variable sizes of M demonstrated intra-segment recombination. 

## 4. Heterologous Recombination in Bacteriophage φ6

Insertion of the *kan* cassette containing the hairpin quickly led the Mindich laboratory to examine heterologous recombination in φ6 [49]. Clear plaque variants derived from φ6K1 were found to carry normal-sized L and S segments but had M segments with sizes varying from a few hundred bp larger than normal to several hundred bp less than normal. Sequence analysis found that the aberrant M segments did not have the 3′ ends of the initial M segment. Of the first three examined, two contained the 3′ end of L, and one had the 3′ segment of S (Figure 2A–D). The crossover-point in all three of the recombinant segments was upstream from the insertion point of the of the *kan* cassette. What proved to be of great interest was the observation that the points of crossover showed little sequence identity, i.e., at most only two to six identical bases (Figure 2B,D,F). Recombination was also observed in an S segment of CS that was derived from φ6K1. The S segment demonstrated an internal deletion of 189 nucleotides within the 3′ noncoding region (Figure 2E,F). The clear plaque variant produced by the carrier culture was a recombinant between segment M and the 3′ end of segment S that contained the internal deletion described above. The final conclusions at this stage of the research proposed that the recombinant segments were extremely unlikely to be the result of homologous recombination. At this point, it was known that entry of two RNA molecules of the same segment type into the PC is, at best, an extremely rare event; therefore, biparental homologous recombination was considered to be exceptionally unlikely.

Several significant conclusions were presented in the discussion of the study results, and these are summarized below.

Heterologous recombination could be a method of repair of damaged RNA genomes. Interestingly the recombination repair compelled two of the three φ6 segments to share identical 3′ ends, yet RNA and viral replication was not affected. The 3′ end sequences, therefore, could in no way define the specificity of packaging of individual segments.

The recombination was responsible for what the authors termed “dramatic changes in viral genomes” exceeding that from homologous recombination.

The recombination took place in the almost total absence of sequence identity with the exchange of heterologous ends. The authors acknowledged that the frequency of recombination could be much higher than the experimental assays showed, as only plaque-forming virus was selected. In addition, the authors acknowledged that the recombinants could have been induced by the nature of the stable hairpin of the *kan* construction that added 1.2 kbp to the 4 kbp genomic segment M.

## 5. RNA Structure Facilitates Recombination

The focus of the next publication described how the RNA structure facilitated recombination and rescue of viral segments. The assays dramatically demonstrated the phenomenon utilizing reporter genes incorporated into large hairpin conformations [50]. The study also utilized the PC assembly and rescue methodology, this time employing four unique methods of altering the m transcripts to verify heterologous recombination.

### 5.1. Kanamycin Sensitivity

First the M segment carrying a *kan* cassette bound by G homopolymer upstream of the kan cassette and C homopolymers downstream was modified to see if the hypothesized hairpin secondary structure promoted the replicative instability. Transcription plasmids were constructed in which one of the homopolymer arms was removed by partial digestion by *PstI* following the progressive digestion with exonuclease *BAL31*. Transcripts were isolated and packaged into the recombinant PC and used to transfect spheroplasts of HB10Y pseudomonads. In contrast to the unstable and heteromorphic plaque-producing phage (such as φ6K1 described above) the removal of one of the homopolymer arms resulted in an entirely stable transducing phage containing the *kan* insert.

### 5.2. Β-Galactosidase Expression in Phage Sensitive Pseudomonad Host Cells

The second assay method utilized a Lac complementation system for pseudomonads that employed a wide-host-range plasmid that carried the ω fragment of the *lacZ* gene, a system analogous to *E. coli* cells containing the *lacZΔM15* gene on the F′ episome, allowing blue–white screening [51]. cDNA copies of segment M containing inserts of the α portion of the *lacZ* gene were constructed in which the insertion was either bordered or not bordered with poly-G on 5′ end or poly-C on 3′ end arms. Figure 3, reproduced from the Onodera et al. (1993) publication, visually shows the results. In the presence of the X-Gal substrate, transducing phages carrying *lacH* gene without one of the homopolymer arms were stable producing only blue plaques, whereas comparable phages but with the homopolymer addition were unstable in that both blue and clear plaques were produced. The HB10Y *lacZ*-based β-galactosidase expression was found to be length-dependent. The shorter *lacG* signal was detectable only with the fluorescent reagent 4-methylumbel-liferyl-β-D-galactoside (MUG) as a fluorogenic substrate for β-galactosidase and was not blue detectable with X-gal. The *lacG*-carrying phage with a hairpin structure produced by two homopolymer arms was so unstable that the virus purification from the plaque was difficult because the vast majority of the plaques were not fluorescent. However, as a rule, increasing instability resulted when the homopolymer arms were extended, regardless of whether *lacG* or *lacH* was used as the reporter gene. The electrophoretic mobility of the viral RNA segments showed that the size of the M segment was altered by heterologous recombination event and yielded phage stocks with normal titers (Figure 4).

### 5.3. 3′ Truncation of the M or S Segments and RNA Repair by Recombination

The 3′ ends of each of the three segments of φ6 have low sequence similarity but are very structurally similar. The region of similarity covers the last 75 nucleotides at the 3′ end, starting with AAGU for all three strands. The predicted secondary structure of the region of similarity is visualized in Figure 5. The identical units are boxed. Even nonidentical nucleotides kept the same secondary structure. The removal of that region of similarity does not affect the packaging ability but prevents minus-strand synthesis. In the third assay method, in vitro transcripts of either the S or the M segments were truncated at the 3′ end at positions 2683 (a *ClaI* restriction site for S segment) and 3830 (a *PstI* restriction site for M) [52]. These modified RNA segments were packaged along with the other two unmodified φ6 genome segments and transfected into host HB10Y cells. The rescue assay provided significantly fewer plaques than expected if wildtype transcripts had been packaged. However, from this limited number of plaques, phage stocks were prepared and the genomic RNA was analyzed; all of the rescued *s* or *m* transcripts were found to be heterologous recombinants. The missing 3′ end of the truncated segment was replaced with a 3′ end from one of the heterologous transcripts (i.e., the donor strand). The crossover points were 5′ to the original truncation site, representing a “repaired” RNA segment. Although the crossover positions were not found to be dependent on significant sequence identity among the transcripts, the possibility could not entirely be ruled out. Therefore, the last of the four methods described in the paper was designed to check off this chance.

Several experiments were next performed to show if homology influenced the recombination events. A sequence duplicated 27 bp upstream of the *ClaI* site on the S segment was inserted into the M segment just upstream of the poly-G arm. The resulted construct produced unstable phages. If the recombination depended on the 27 bp identity or at least preferably occurred at the homology site, all of the crossovers would be of a predicted size and sequence. The predicted unique size was not observed, and, when the M recombinants were analyzed by sequencing, there was no evidence of crossovers in the repeated region, showing that homology did not facilitate the recombination.

The 1993 studies concluded by providing a model of φ6 m segment heterologous recombination via a copy choice mechanism (Figure 6). The model explains how the 5′ end blocked (or truncated) plus sense mRNA obtains a replacement 5′ end from one of the other two complete genome transcripts. Although the model illustrates the potential mechanisms using the m segment as a target, the assumption was made that the other two strands would undergo recombination using the same copy-choice process. Referring to Figure 6, the s segment commences minus-strand synthesis in a normal way but initiation of synthesis is blocked on m. Two pathways termed A and B can potentially repair the m- segment. Pathway A can be explained as follows: the new 3′ end is derived from the l segment when it is used as a template, and the continuation of synthesis shifts to the m plus strand. L synthesis then commences once again and continues to completion. Pathway B is more complex in that transcription of the new plus strands displaces the chimeric minus strand from its original m template. The original template would then reinitiate minus-strand synthesis, which proceeds normally. The B pathway is less likely based on a previous observation that minus-strand synthesis must start and complete on l before any transcription takes place [53]. However, it was clear in the study that *both* s and l can donate 3′ ends, and, if the completion of l transcription is not a hard and fast rule, then this pathway works to resolve the complete dsRNA genome.

## 6. The Secondary Structure at the 3′ Ends of the Plus-Sense RNA Segments Required for Minus-Strand Synthesis

Approximately another year passed until the next paper in the recombination series was published [52]. The study focused on the consequences of removal (or modification) of the secondary structure at the 3′ ends of the three genomic plus sense segments responsible for minus-strand replication initiation [49]. The aim of the modifications was interruption or alteration of the secondary structure predicted at the segment 3′ ends. An initial pilot experiment was performed entirely in vitro using l, m, and s transcripts with truncation at the 3′ end designed to show that, while packaging could occur, the truncated transcript did not support its own minus-strand synthesis. Figure 7 shows the effect of single-nucleotide deletion in the M cDNA sequence, along with the calculated secondary structure. According to the secondary structure predicted using the *forna* online RNA secondary structure visualization tool (http://rna.tbi.univie.ac.at/forna/, accessed on 17 August 2022), the changes in structure could be divided into two groups. In the first one, shown in Figure 7B, arm I was altered. Structural changes in that arm caused recombinations in the many, but not all M segments of progeny phages, and the replication rate of M segment was approximately the same as that of unmutated viruses. In the second group, the deletions caused a shortening of the arm II (Figure 7C). These mutations caused a significant reduction in the M replication rate (<0.5), and all the produced phages were recombinant. One control deletion was placed at a segment position significantly upstream of the putative hairpin region and did not cause any recombination events in the m segment and did not affect the replication rate of the m segment. In a surprising result, the deletions did not always significantly alter the rate of minus-strand synthesis as measured by radioisotope incorporation with autoradiogram analysis. However, the minus-strand synthesis results did not correlate with actual plaque formation after spheroplast transfection. Therefore, the virus viability was far more sensitive to 3′ end alteration than minus-strand synthesis.

The concluding remarks in this paper refined and tightened the heterologous recombination model presented earlier where lost or diminished minus-strand synthesis promoted heterologous recombination. In this 1994 publication, minus-strand synthesis was often observed to remain normal (i.e., estimated by autoradiogram observation) in situations when recombination still occurred. Presented with this seeming contradictory evidence, the adjusted model proposed two components:(1)Minus-strand synthesis normally begins in the three φ6 transcripts simultaneously within the confines of the PC. If one of the three is undergoing synthesis at a reduced (or zero) rate, the polymerase responds by switching to a functional and adjacent 3′ end, resulting in copy choice-based recombination. This notion posits that RNA segments that are even slightly impaired as templates could also induce recombination.(2)The secondary structure at the 3′ ends of the genomic segments guard against nucleases that would degrade the plus-sense RNA. Since it is known that the 5′ ends contain the packaging recognition sequences, 3′ end deleted (or damaged) RNAs would still be packaged but minus-strand synthesis could not commence. The heterologous recombination is a method of RNA repair that allows the virus replication cycle to continue.

Lastly, the paper concludes by stating the possibility that the recombination noted in φ6 might be extremely different from that seen in other RNA viruses because the dsRNA synthesis occurs in the strict confines of the PC. A viroplasm-based replication mechanism could be “looser” with the genomic RNA not in such close proximity. This concept is further addressed in Section 13). Additional refinement of the model, in a subsequent paper, awaited an additional year after continued research efforts.

## 7. The Serial Dependence of Packaging of the Individual Segments

In tangent to the studies under discussion in this review, the next paper published in May 1995 (while not specifically a recombination paper) needs to be included in the entire story as it described conditions that govern the packaging mechanism. Crucial as well to the continuing studies was the introduction of the ribonuclease protection technique for the rapid detection of transcript packaging [54]. Once packaged, the P^32^-labeled RNA was protected from RNase I digestion. The part of RNA left outside from PC was RNase I-sensitive and was easily digested. The serial dependence of the mechanism was described in that a *strict order* was observed where s is first packaged followed by m, and then finally the l transcript. This observation superseded previous studies (that included one author of this review, P.G.), which concluded that segments were packaged individually [55], while the laboratory of Dennis Bamford, at the University of Helsinki, found that s and m are packed individually but l is dependent on m [56]. These two assumptions did not take into consideration the extreme packaging sensitivity to Mg^2+^ ions in the presence of 1.5 mM phosphate and was also ATP-dependent. The independent in vitro packaging of each segment occurs at magnesium concentrations above 6 mM (used by P.G.) and high salt-based isolation of the recombinant PC (used by D.B.), which led to the initial assumptions of packaging independence. At 10 mM Mg^2+^, even nonspecific control RNA (lacking any φ6 sequence) was packaged.

Most notably, the discussion section of the paper established a detailed model of the packaging of the three transcript segments that was later confirmed by structural analysis of the PC components [23,57]. The genomic segment entrance into the particle is ATP-dependent, and the segment selectivity is intrinsic to the internal PC structure. At the low magnesium concentration (estimated at 4 mM in bacterial cells), only when the s segment is packaged can m bind to the PC and subsequently be packaged. The PC then has the ability to bind the l segment which is next and finally packaged. The completely packaged PC can then commence negative-strand RNA synthesis to produce the mature dsRNA genome.

## 8. The Union of Two Mechanisms: Packaging and Heterologous Recombination

September 1995 saw the next publication in the series, in which the packaging and heterologous recombination mechanisms were firmly linked [58]. The experimental protocol made use of a hairpin structure consisting of a *lacH* gene flanked both sides by poly-G and poly-C homopolymer arms inserted as a large loop within the 3′ region of the s and m transcripts. One control construction based on the s segment had a strong hairpin but lacked the *lacH* loop near the 3′ transcript end, while other controls contained *lacH* but lacked the GC boundaries. The RNase I protection assay was utilized to analyze both packaged and partially packaged RNA products. The major observation was straightforward in that *lacH* constructs were RNase I-insensitive, indicating that successful packaging had taken place. The RNA constructions that contained *lacH* bounded by the poly-G and poly-C arms were only partially protected from the ribonuclease, and the size as measured by gel electrophoresis indicated that only the hairpin loop portion was RNase I-sensitive. Essentially, the hairpin loop formed a packaging blockage point where only the distal 5′ could enter the PC. The hairpin consisting of homopolymeric GC borders but lacking the *lacH* loop was also digested along with the portion 3′ from it. While the study only utilized an in vitro assay, the m hairpin construct utilized was the same one that yielded a recombinant bacteriophage, as described above (Section 6), linking the two phenomena.

The results of this 1995 study can be summed up in the schematic models presented in Figure 8 reproduced from the publication. Two distinct models were suggested to describe the findings, of which the first had strong experimental proof.

(1)A large and strong hairpin cannot be taken through the packaging portal of the PC leaving the 3‘ end of the transcript stuck on the outside. Only the portion of the transcript segment beyond the hairpin would be packaged, and the 3′ end remaining outside the PC would be subjected to RNase I digestion.(2)A second possibility suggested that the entire transcript is packaged in spite of the added large secondary structure, but the viral RNA polymerase is incapable of pushing through the strong hairpin.

In either of the models, the minus-strand synthesis could switch to the 5′ side of the hairpin, bypassing the RNA synthesis block. The results noted by the size determination on agarose gels of the digested RNA supported the first model. Addition support was obtained for the observation that ssRNA packaging proceeds from the 5′ end since it was the 5′ end that was protected from RNAse digestion.

Packaging of the s segment was incomplete because of the hairpin blockage, but the m segment still entered, arguing that more than one entry portal was available—a concept not entirely obvious as the specific packaging order implied only one active portal, as in DNA bacteriophage. The one active portal compels a mechanism where only one of the P4 hexamers differentially binds to the P1 framework, producing a unique site on the PC where packaging commences and continues [59]. This controversy has still not been resolved. While the precise dimensions of the hairpin loop structure were not reported and the P4 hexamer atomic structure delineation was still in the future [60], the authors of this paper predicted that the locked hairpin structure could be visualized by electron microscopic techniques—a proposal that remains worthy of trial in the present.

## 9. Acquisition of Additional RNA Segments by φ6 and Heterologous Recombination

The Mindich laboratory published two back-to-back companion studies in 1995 [61,62]; both were initially written in June 1995 and released for publication after the paper described in Section 8. These two papers are summarized together in this section of the review. The two references are recorded in this review according to the page order of the journal but can be read as either first without loss of any comprehension. Together, the publications demonstrated that multiple copies of one of the genome segments could be packaged if the segment was of a reduced size and heterologous recombination occurred.

We choose to first summarize Mindich et al. (1995) [62] as it is conceptionally simpler and demonstrated in vitro that multiple copies of a given genome segment could be packaged but only if it was diminished in size. This study made use of recombinant cDNA encoding transcripts containing significant size deletions. The relative stoichiometry of the packaged shorter segments was determined by (1) visual estimation after agarose gel electrophoresis and autoradiography of isotope-labeled RNA, and (2) direct quantitation by scintillation counts of stained and isotope-labeled bands removed from agarose gels. To illustrate the methodology and the quantitative estimation, we summarize one example, the l segment and s segment packaging ratios. The radioactive count ratio of wt segment l (6374 bases) to wt segment s (2948 bases) should be 2.16. When segment l with deletion (only 1552 bases) was packaged, the ratio of 0.52 was predicted on the basis of the segment sizes. However, the observed ratio was 2.28, which is a value close to the wt l segment. Therefore, four deleted l segments were actually packaged maintaining the l genome quantity per capsid. Similar assays were performed to establish ratios of the other two deletion-containing segments, and multiple genome packaging was confirmed. The results of the in vitro analysis strongly suggested that more than one representative of a specific genome type can be packaged into a PC if the segment size is substantially less than the wt. The authors proposed that a “bin” for each segment must be filled even if it requires more than one unit of the segment (what they described as a quota for each segment). The model required multiple binding sites for each genome segment and a sensor mechanism for the amount of each segment.

The next study by Onodera et al. [62], which is conceptionally more complex, requires greater explanation. The recombination assays relied on the construction and ssRNA genetic acquisition of cDNA copies of partial segments of m and l in recombinant plasmids that replicated and were each expressible in the φ6 pseudomonad host cells. The assays required the identification and quantification of the incorporated segments. The crucial constructions are illustrated in Figure 9A, which is based on Onodera et al. (1995) [62], schematically showing each of the cDNA recombinants. The s and m segments were expressed in the wildtype form, as noted on the figure. The initial experiment used the construction of pLM1009 that encoded only wildtype gene 1 along with the normal packaging signal and 3′ end of transcript l. However, genes normally located on the L dsRNA segment, 14, 7, 2, and 4, were removed from the recombinant cDNA. The deleted transcript was packaged into an in vitro assembled NC with along with a mixture of the other three transcripts (i.e., s, m, and l). The combination of transcripts assembled into the NC *included* an additional and *complete* l transcript from φ6 *sus351* that contained a nonsense mutation in gene 1, rendering it unexpressed [63]. Bacteriophage plaques were obtained after spheroplast transfection of host cells carrying plasmid pLM1003 (Figure 9B), which contained genes 14, 7, 2, and 4. A representative bacteriophage particle was then selected, designated φ1980, and the dsRNA L segment was found to be missing nucleotides 392–3588 (i.e., missing genes 14, 7, 2, and 4), just as in the pLM1009 construction (Figure 9C). The four-segment bacteriophage tended to delete the fourth genome when replicating under the influence of plasmid complementation. The PC probably remained unchanged and could not be further expanded to accommodate the extra genome (L from φ6 sus351) and could only inefficiently package the four genome segments, i.e., at low frequency.

Furthermore, the stoichiometric quantity of the packed L segment with deletion (LΔ) was estimated approximately twice that of the S and M genome segments, suggesting that at least two copies were packaged along with the expected one copy of S and M. This bacteriophage was found to propagate on a host cell that carried a plasmid pLM1003 which contained genes 14, 7, 2, and 4 and the pUC8 multiple cloning site (mcs) situated at the cDNA 3′ end. These recovered bacteriophages were next plated on wildtype host cells lacking plasmid pLM1003 with plaques obtained only at an extremely low frequency (estimated at 10^−8^, Figure 9D). Nevertheless, in spite of the low viral recovery, agarose gel size measurement of the dsRNA genome showed that a new segment had indeed been acquired and was received from the complementing plasmid pLM1003 (Figure 9C, columns 3–5). Of interest was that the new segment had a normal 5′ end, although the transcript from the plasmid-borne recombinant insert would not be expected to start at this position. Therefore, it was evident that in vivo trimming had occurred to create the normal packaging signal expected to be on the viral genome. In addition, the 3′ end of the s or m transcript was present as the ends were the result of heterologous recombination. Donor transcripts from pLM1003 can include mcs in the final recombinant product. Sequence analysis indicated extremely limited runs of sequence identity at the or in the vicinity of the crossover site (Figure 9E).

The chimeric plasmid constructed with 307 bp of packaging sequence of M segment, genes 2, 4, and 7, and the replication sequence from L could effectively replace pLM1003 in terms of providing the missing 2, 4, and 7 genes. The resulting recombinants can either bear the four dsRNA segments (LΔ, strand with genes 2, 4, and 7, S, and M) or three dsRNA segments, but the M segment incorporates the 2, 4, and 7 genes and becomes significantly larger in size.

The study authors emphasized that the ability to package the additional segment was likely unrelated to a mutational change within the procapsid. Plating a “duplicated segment” bacteriophage on host cells carrying a complementing plasmid allowed a high number of plaques to form on the order of several thousand. However, when the host cell lacked the complementing plasmid, only a few plaques were seen. This ability of phages carrying the gene deletion in one of the segments propagated in host cells, with complementary plasmids then also generated for the M segment. 

In summary, the major conclusions from the companion studies can be summarized in four points:(1)Multiple copies of the genome segments of the same specificities could be packaged into a PC if the sizes were smaller than normal, while the total amount of RNA was not significantly changed. Mutations making the capsid permissible for multiple transcript packaging was unlikely but could not be entirely ruled out.(2)Heterologous recombination was seen to occur at the 5′ ends of the segments, while the recombination observed was previously limited only to the 3′ ends. Of great interest (even to current viral studies) was the realization that the recombination was likely to take place at any region or point throughout the entire length of the segment. The 5′ crossovers occasionally showed limited sequence identity, suggesting a different mechanism from the 3′ event.(3)The reverse genetic method was expanded in the study as it was based on segment acquisition from propagating bacteriophages in host cells carrying plasmids that transcribed complemented genomic segments, i.e., via acquisition vs. spheroplast transfection of in vitro packaged PCs.(4)In the course of altering the l segment, it was found that gene 1 could be moved from the 3′ to the 5′ end of the segment without loss of function, indicating that the cystovirus gene positions need not be fixed for proper function. When genes 7, 2, and 4 were located at the 3′ end of the m segment, the virus remained viable, a surprising observation as the three genes (normally on the l transcript) were considered to function only in early protein expression. The alteration of gene position on segment l was later found to occur naturally in bacteriophage φ8 when it was isolated and classified by the Mindich research group. In φ8, the gene for an ortholog of protein P7 was noted to be located at the 3′ end of the plus strand rather than near the 5′ end [64].

## 10. An In Vitro System for the Investigation of Heterologous RNA Recombination

The next study in the recombination series, published in 1997, described the development of an in vitro system used to investigate heterologous recombination [65]. The developed assay used all previously successfully applied methods and forced the recombination event to happen in the recipient m strand through deletion of the 3′ end and gene P3. Although the assays produced recombinant plaques, recombination took place within the PC during minus-strand synthesis. Several recovered recombinant plaques were analyzed to determine the crossover points in M, S, and L segments. The launch positions for both l and s segments were found to be broadly distributed and not originating from any specific site on the genome segment. Yet the landing points on segment m located in the middle of the RNA strand and few other landing points were near the 3′ end in the vicinity of the deletion of the replication initiation signal. Most of the recombinant m segments were smaller than wildtype m. As in the previous studies, the crossover regions showed limited sequence identity.

Examination of genome recombinants showed that the crossover regions frequently had gaps between the point of crossover and a limited (but identifiable) upstream identity. The gap sequence was usually that of the recipient strand. The authors of the study then reported that, in 17 examples of this second set of recombinant viruses isolated, they saw an upstream identity of two or more bases within five bases of the crossover position. In 12 examples where two regions of identity were separated by *only two* bases, the intervening sequence was that of the target m segment, and the suggestion was made that the donor strand is trimmed back after landing on the target template. 

The study demonstrated that a recombination assay could be established in vitro by exploiting the packaging and RNA synthesis abilities of the φ6 PC. The major conclusions and observations of the study were as follows:(1)The distribution of the launching and receiving points for recombination of the viral RNA had no specific spots for the process to occur.(2)The φ6 recombination became possible with at most only a limited region of similarity at the crossover point. The authors hypothesized that, in other positive-strand RNA viruses (i.e., poliovirus and brome mosaic virus) the association between recombining RNA could require mediation by complementary sequences as, within the viroplasm, a “low” RNA concentration could limit this process. In cystoviruses, packaged RNA is forced into high density; therefore, complementary sequences are not a recombination requirement.(3)Heterologous recombination appears to take place during minus-strand synthesis, and the authors described a revised model of heterologous recombination facilitated by only limited regions of complementarity. Summarizing the model, they described the following:
a.A nascent minus strand is displaced from its template along with the viral RNA polymerase landing on another of the two templates. The polymerase would then continue, extending the nascent strand on this new template.b.The nascent strand could be entirely or only partially single-stranded. The 5′ end can still bond with its *original* template, but the 3′ end (leading edge) is single-stranded. The leading edge of the nascent RNA chain can anneal to the new template at a region of limited sequence identity.c.The results described in the paper suggested that there is frequently additional identity upstream of the crossover point, but the intervening sequence favors the recipient strand. The authors also suggested that the leading edge of the of the nascent transcripts can be truncated back by digestion prior to synthesis, accounting for the intermittent and limited regions of sequence identity in the vicinity of the crossover region.

An implication of the model is that the RNA polymerase is mobile and not locked into one position within the PC, allowing its transit along with the nascent strands. Later work using cryo-electron tomography and single-particle reconstructions showed that this is indeed the case, and that not all available viral RNA polymerase (i.e., P2) positions are occupied in the PC [54,55].

## 11. Reverse Genetics and Recombination Established in the φ8 Bacteriophage

The next paper in the recombination series, published in 2001, expanded the study of reverse genetics and recombination to φ8 [18,66,67]. φ8 is most distant from φ6 with regard to amino-acid sequence, as well as genome organization and structure; in particular, the P8 proteins do not form the shell around the PC, but incorporate into the viral envelope along with the envelope proteins P3a, P3b, PF, and P10 [62,68]. The φ8 genome rescue was based upon the ability of the virus to acquire plasmid transcripts from infected cells, incorporating these into its genome as substitutions for defective genomic segments. Therefore, segment recombination was facilitated using the acquisition mechanism. Significantly, φ8 virus was found to undergo homologous and heterologous segment recombination in contrast to mostly heterologous recombination in φ6. Therefore, the initial aim of the study was to analyze a bacteriophage variant that could recombine with plasmid transcripts in a manner allowing homologous vs. heterologous recombination to be differentiated. The five observations and methods are explained below.

(1)*Preparation of the tester bacteriophage*: In order to produce a tester bacteriophage, the following procedure was employed: first, m sequence cDNA was modified by deleting the 3′ terminal genes F and G, and lac α was inserted after gene 3b (Figure 10). The modified m sequence was placed in an *E. coli* to *P. syringae* shuttle vector, and this recombinant plasmid was transformed into a host for φ8 that was designated LM2489. The next step was to design a bacteriophage that carried the kan gene in place of gene 3. This bacteriophage (selected by plasmid-based transcript acquisition) was designated φ2745 and was able to replicate in host strains that expressed genes 3a and 3b from a recombinant plasmid (this recombinant was also derived from a shuttle vector). φ2745 was able to replicate in a host strain carrying the pLM2764 plasmid and transit through this additional strain, allowing selection of yet another tester bacteriophage where genes F and G were replaced by lac α, designated both tester phage 21 and φ2756 in the original paper, which can cause confusion. Phage 21 (i.e., φ2756) had the ability to propagate on hosts that expressed genes F and G from a recombinant plasmid and could acquire the plasmid-produced transcripts.(2)*Homologous recombination in φ8*: Several cDNA fragments were synthesized that had the 3′ end of segment m with the genes F and G but only portions of gene 3b (Figure 10). The 3b sequences each were of variable size in that they started within the gene and ranged out to the gene end, i.e., a series of overlaps with the bacteriophage gene, *producing a homologous target*. The fragments were ligated to an *E. coli/P. syringae* shuttle vector and transformed into host cells permissive for φ8 replication. Since the transcript lacked the pac sequence, it could not be packaged into bacteriophage particles and was only capable of complementation with the FG genes. φ2756 was utilized as the infecting particle, and numerous progeny bacteriophages were isolated that could infect host cells lacking any of the complementing plasmids. The set of *plasmid-independent* progeny bacteriophages had both abnormal-sized and normal-sized m segments as judged by agarose gel electrophoresis. Sequencing of RT-PCR cDNA of the normal-sized m segments demonstrated that they had a wildtype sequence, showing that the recombination was homologous.(3)*Heterologous recombination in φ8*: In order to prepare a target for heterologous recombination, the m segment was modified to accommodate the gene for kanamycin resistance within gene G. The recombinant sacrificed the G gene product, but the study authors reported that the gene loss or truncation did not prevent bacteriophage replication. The recombinant m segment was provided to WT bacteriophage via the transcript acquisition method, and carrier-state progeny were selected on kanamycin-containing media as in Onodera et al. (1992) [45], described in Section 3. Supernatant fluid of cultures derived from kanamycin-resistant colonies yielded bacteriophages that proved unstable with regard to kanamycin resistance. These bacteriophages were plated on host cells that provided complementary proteins F and G (produced from recombinant plasmids), where the *kan* gene was frequently lost by template switching, producing m segments with 3′ ends of the l or s segment. The heterologous recombination was similar to that observed in φ6 in that crossover positions only had two to three bases in common, similar to that described by Qiao et al. (1997) [65] and as reviewed in Section 10.(4)*The role of pac sequences influences recombination*: The next goal required to complete the φ8 recombination assay was the construction of a set of plasmids that each encoded a transcript of a portion of the m segment that encoded parts of gene 3b. The plasmids encoding the m transcripts were able to complement the defect in φ2756, and, when the transcripts carried *pac* sequences (derived from the s or m segment), acquisition was extremely efficient. When the *pac* sequence was missing, the opposite result was observed, i.e., extremely low acquisition. However, the authors noted that, regardless of the presence or absence of *pac* (and its significant effect on acquisition), the ratio of heterologous to homologous recombination was the same.(5)*The role of the 3′ replication initiation site in transcript acquisition*: Plasmid transcripts that lacked both a 5′ end pac sequence and a 3′ end replication initiation sequence could only complement bacteriophage deletions but could not be acquired.

The primary and most significant contribution from this study was the presentation of a model for the mechanism of template switching-based recombination, as shown in Figure 11. The model was presented at roughly the same time that the atomic structure of the φ6 RNA polymerase was published [58] and included the observation that the RNA template pathway through the protein is entirely enclosed, having implications for facilitating template switching. The polymerase was seen to be completely recessive as the 3′ end of the template enters the polymerase. Notably, the atomic structure ultimately illustrated that chain initiation is consistent with the elongation template switching/recombination model presented in this study. Figure 11 details the model.

In part “A”, the template positive-sense RNA inserts the 3′ end within the entrance pore of the polymerase, and NTPs enter from another pore. Minus-strand synthesis is initiated in “B” and leaves through the exit pore. “C” shows the minus strand in a displacement phase which can provide a free 3′ end. Position “D” of the figure represents another template plus strand that lacks a 3′ end containing the polymerization initiation sequence. This second and deleted template is free to move through the polymerase and partially leave through the exit pore. In “E”, the displaced strand anneals to the free (but 3′ end deleted) second template, whereas, in “F”, it primes minus strand polymerization. The major points to emphasize are the following: (1) the deleted template, despite entering the catalytic site, is not locked in position and is free to move (referred to as “scanning”) back and forth; (2) after minus-strand annealing to the positive strand the template can move back into the catalytic site, and minus-strand synthesis is initiated. The model allows both heterologous and homologous recombination.,

## 12. Construction of Carrier-State Viruses with Partial Genomes of the Segmented dsRNA Bacteriophages and the Establishment of the Plasmid Electroporation Rescue Method

The final report in the series of studies introduced a new methodology to rescue cystoviruses φ6 and φ8 from genetic material and isolate viral recombinants [69]. Despite not specifically a study examining RNA recombination, it warrants inclusion and mention in this review for the introduction of a methodology for the rapid manipulation and rescue of the cystovirus genomes utilizing a simpler procedure than the nucleocapsid assembly system. The method required the electroporation of plasmids that each carried a cDNA copy of a genome segment into pseudomonad host cells that expressed either the SP6 or T7 RNA polymerase. Therefore, any cDNA copy to be expressed had to be placed in a recombinant plasmid after an SP6 or T7 promoter. Significantly, these expression plasmids were ColE1-derived and lacked the ability to replicate in the pseudomonad host and the carrier state, and expression of recombinant markers was simply launched by the vectors.

As in earlier studies, the *kan* resistance gene was placed into select gene segments of φ6 and φ8; selection was for antibiotic resistance, and this marker gene was either inserted into noncoding regions or replaced one or more viral genes. The electroporation of all three cDNA-carrying plasmids allowed efficient selection of kanamycin-resistant *P. syringae* colonies. An unexpected observation showed that insertion of cDNA copies of only s and l could produce antibiotic-resistant colonies at high efficiency. The combination of l and m was less efficient in the carrier production on φ6 and did not work at all in φ8. An additional surprising observation noted that carrier bacteriophages that contained only two segments had an equal number of segments. This counterintuitive discovery appears to contradict the head-filling mechanism described in recombinant φ6 that packaged only partial genomes. The study authors realized that the unusual l and m combination required a mutation in gene 1 that encodes protein P1. The Mindich laboratory previously demonstrated that, in φ6, similar genome constellations were a consequence of changes in the P1 amino-acid sequence [70]. RNA was isolated from two of the l and m φ6 carrier-state cultures and l cDNA synthesized. Electroporation of plasmids carrying the new l sequence into *P. syingae* along with a cDNA copy of m carrying the *kan* resistant gene gave a high frequency of carrier-state colonies. The two new l gene sequences were determined, and amino-acid changes were found in P1 that mutated (1) a glutamate to a lysine residue, and (2) a threonine to a proline residue. The φ8 carrier state differed as it depended on the presence of segment s, and the study concluded with examination of this segment in order to determine which genes were required for carrier-state maintenance. The study reported that genes 5 and 9, and ORFs I and J were successfully sacrificed by substitution with the *kan* insert. However, deletions of any portions of genes 8 and 12 prevented the formation of the carrier state. This result is of the greatest interest as protein P12 is nonstructural and facilitates viral envelope assembly, while P8 does not form a nucleocapsid matrix in φ8; therefore, it remains unclear why the proteins are crucial to assemble carrier viral particles.

In summary, this final paper described in the review reported the following significant observations which suggest future study:(1)The carrier state was established without the selection of viral mutations as kanamycin selection by itself was adequate. Although the data were not shown in the paper, carrier-state stability was also said to increase with mutations that lowered the viral polymerase activity.(2)Neither φ6 nor φ8 can dispense with packaging s in absence of a suppressor mutation supporting the model where RNA binding sites for m and l are only revealed when s is packaged [71]. At the time of publication, the mutations that modified the binding order were noted to be localized in the distal region of the linear sequence of gene 1. Although the details are beyond the scope of this recombination review, the later description of the P1 protein structure at atomic resolution published in 2013 showed the specific locations of the amino acids that affect RNA packaging. The subunit is a flattened trapezoid that displays two conformations that have intermediate states of maturation, allowing the sequential packaging of the three genome segments. The mutations that sacrificed s binding were seen on the PC exterior [24,72].(3)φ8 packaging appeared to require genes located on the s segment as described above. In particular, the results indicated that the specific function of the morphogenic protein P12 is incompletely defined in that it seems to have packaging activity separate from its envelope acquisition property demonstrated in φ6 [73]. The function of P8 (which assembles the icosahedral matrix of the other cystovirus types) is not defined in φ8, in that it was seen to be required for packaging and possibly for transcription.(4)Lastly, the paper concluded with the suggestion to utilize a carrier system to analyze *Reoviridae* infection in eukaryotic cells. Rescue has not been accomplished for any of the reoviruses, and the carrier method offers an alternative methodology to study the virosome. The proposal is likely applicable to other viral systems, and this will be addressed in the next section.

## 13. Summary and Conclusions

The Lai (1992) book chapter referenced at the beginning of this review [4] concluded with a challenge to demonstrate the possible occurrence of genetic recombination in other RNA viruses, and the work summarized above clearly made a major contribution to this calling. Overall, the cystovirus studies demonstrated that a controlled experimental approach to RNA virus intramolecular recombination was achievable. The prevalence of heterologous recombination as opposed to homologous recombination was a notable observation and would suggest that, in the wild, viral RNA genomes could be exceptionally labile. For example, a peculiar, but perhaps not all that rare, bat-derived coronavirus recombinant (Ro-BatCoV GCCDC1), which integrated via heterologous recombination a P10 syncytial-forming protein from a bat orthoreovirus, has been described [74]. Therefore, as an example, with analogy to the bacteriophage history described above, a brief discussion of a betacoronavirus recombination by copy choice in SARS-CoV-2 presents a valuable contemporary lesson. Specifically, an acquisition mechanism of the furin endoproteolytic cleavage site and the human angiotensin-converting enzyme 2 (hACE2) receptor spike glycoprotein trimer was described by Gallaher (2020) [75]. As proposed in the study, the SARS-CoV-2 genome is organized as “*functional blocks of RNA information demarcated by short RNA breakpoint sequences that promote recombination at specific nonrandom locations within the viral genome*”. The bat species *Rhinolophus affin*is (Bat-RaTG13) was found to be 96% identical to SARS-CoV-2 on an RNA level but carries an insert encoding four amino acids that are recognized by the endoproteolytic host enzyme furin, i.e., PRRAR (Figure 12) [76]. The entrance of this sequence into the bat sequence has become a major source of the “gain of function” discussion and controversy including the implication that purposeful modification or improper laboratory practice was responsible [77]. The SARS-CoV-2 evolutionary history likely includes positions of breakpoints in the RNA genome that facilitate copy choice jumps by the RNA polymerase complex. A tandem duplication is found before the furin inset sequence, suggesting that the polymerase complex is hindered in its motion, and the stutter could cause a tandem duplication or insertion of extra sequence. The idea is analogous to the scanning and back-and-forth model presented in the Mindich model described in Section 11, which accounts for both heterologous and homologous recombination in the cystoviruses. The coronavirus describes polymerase hindrance as influenced by a base-paired stem-loop structure that a helicase would need to melt out before the polymerase proceeds, a theme well described throughout the cystovirus studies. Indeed, a secondary structure analysis of the SARS-CoV-2 structure identified many such regions in the genome capable of causing polymerase blockages and copy choice recombination [78]. As reported in the Gallaher study [75], significant changes in the sequences of coronavirus types can indicate breakpoints for recombination, and an example was described among ancestors of RaTG13, SARS-CoV-2, and a betacoronavirus isolated from a Malayan pangolin (Pan_SL-CoV-2). The recombination event presumably placed the receptor binding domain for hACE2 into the current SARS-CoV-2. The argument can be made that the cystovirus recombination only occurred within the low-volume confines of the PC structure; therefore, coronavirus recombination must be mechanistically unrelated. However, cryo-electron tomography structural analysis of SARS-CoV-2 particles assembling in host cells indicated that dsRNA filaments were prevalent within the confines of a double membrane, forming a tight replication complex [79].

To conclude, it stands to reason that the ultimate understanding of viral genomic RNA recombination mechanisms will be dependent on the entire delineation of the molecular architecture and geometry of the RNA as it enters and packages within the mature virion particle. The description of the cystovirus genome as a liquid crystalline array in a highly compacted form represents a start in order to complete the goal [41], and significant progress has also been made in describing coronavirus packaging [79]. Albeit entirely laboratory-based assays, the cystovirus methodologies developed by the Mindich studies offer a detailed guide to an inventive direction that should be revisited by current researchers considering examining intragenomic RNA recombination in viral pathogens. At the very least, identification of genomic “hotspots” for viral RNA polymerase copy choice action might have predictive value for potential sites of genome additions and deletions in the natural world.

## Figures and Tables

**Figure 1 viruses-14-02589-f001:**
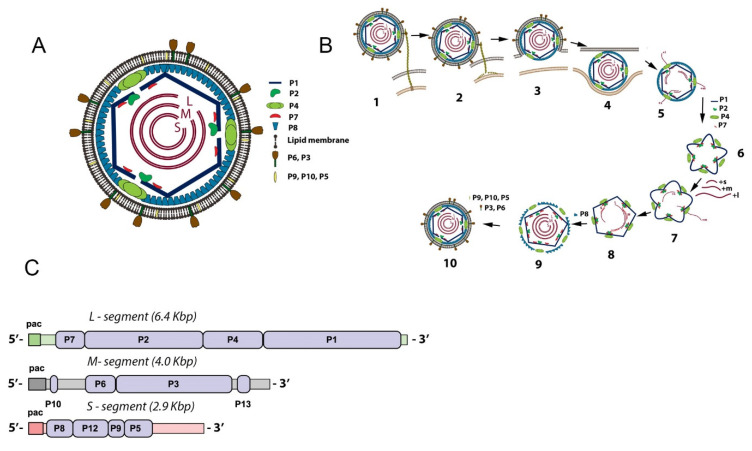
Schematic organization (**A**), replication cycle (**B**), and genome organization (**C**) of the φ6 virion particle. (**A**) The inner layer, procapsid (PC), includes the shell composed of 120 copies of the P1 protein or 60 nonsymmetric dimers of P1A/P1B. Inside the shell are proteins P2 and P7 localized at the fivefold axis of symmetry portal and three double-stranded RNA segments (dsRNA) which have low-symmetry quasi-concentric shell organization. The P4 proteins form a hexameric ring around the fivefold axis. The nucleocapsid (NC) includes the PC and a matrix composed of protein P8. The completed virion has an envelope derived from the cellular bilayer lipid membrane and incorporates proteins P3, P6, P9, and P5 randomly distributed. (**B**) The replication cycle initiates when P3 protein attaches to the host pili followed by pili retraction (1–2). The viral envelope fuses to the host outer membrane (3). The NC enters the periplasmic space and the cytoplasm (4). The P8 matrix disassembles, and transcription begins (5). The PC self-assembles from P1, P2, P4, and P7 proteins (6) and packages the three genome transcripts accompanied by PC expansion (7). ssRNA is replicated to dsRNA (8), and the P8 matrix loosely assembles around the filled PC (9). The cell-derived bilipid membrane is placed around the NC particle mediated by the nonstructural protein P12 (10). This step is followed by cell lysis and virion release. (**C**) The three ds-RNA genome segments and genes encoding the viral proteins (shown in light-blue color). The packaging *pac* signal for each strand located at the 5′ end of ds-RNA is visualized as colored rectangles. The replication signal is located at the 3′ end.

**Figure 2 viruses-14-02589-f002:**
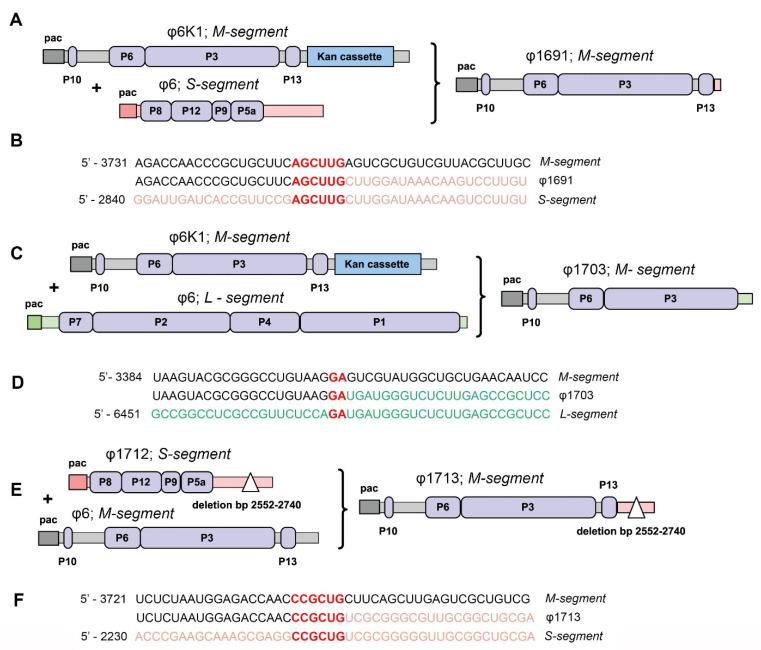
Recombination events facilitated by introduction of kan cassette structurally organized as a hairpin loop. The φ6K1 recombinant virus contains the M-segment with a kan cassette. Isolation of clear plaques from unstable transducing virus confirmed the recombination between M and S or L segments (**A**,**C**). This stable isolated virus φ1697 (**A**,**B**) did not have the kan cassette in the M-segment. The 3′ end of M-segment of φ6K1 (3754–5303 bp) was recombined heterologically with the 3′ end of the S segment (2858–2948 bp). Only 6 bp of similarity existed between two segments. The φ 1703 (**C**,**D**) virus did not have a kan cassette, as did φ1697, and lacked the P13 gene. The φ1703 is a recombination product between the 3′ end of the M-segment of φ6K1 and L-segment of wildtype φ6. There were only two identical base pairs observed. Recombination was also observed in an S segment of CS that was derived from φ6K1 (**E**,**F**). The S segment demonstrated an internal deletion of 189 nucleotides within the 3′ noncoding region. The segment was incorporated into φ1712. The clear plaque variant φ1713 produced by the carrier culture was a recombinant between the segment M from native φ6 and the 3′ end of segment S from φ1712 that contained the internal deletion described above. Only six identical base pairs were observed. Based on Mindich et al. (1992).

**Figure 3 viruses-14-02589-f003:**
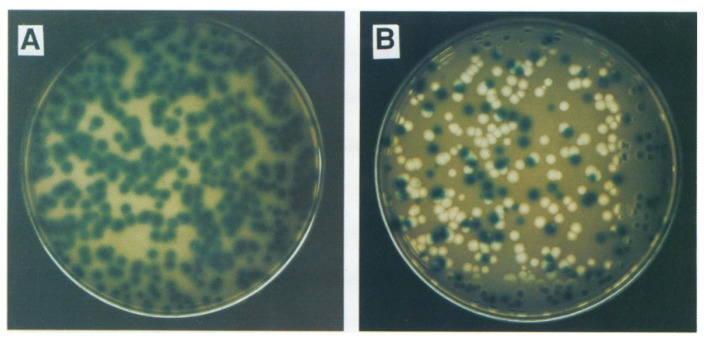
Effect of the hairpin secondary structure in noncoding region of M segment on the stability and homogeneity of the virus. Photograph of plaques from bacteriophage isolates where lacH is inserted in the noncoding region of the M segment and bordered with poly-G and poly-C homopolymer arms (**B**) or lacking one of the arms (**A**). Plaques lacking the one of the homopolymer arms are stable and form uniform blue plaques on LB X-Gal substrate (**A**). The lacH gene bordered by homopolymer arms is unstable, producing the white and blue mixed plaque population (**B**). Reproduced from Onodera et al. (1993).

**Figure 4 viruses-14-02589-f004:**
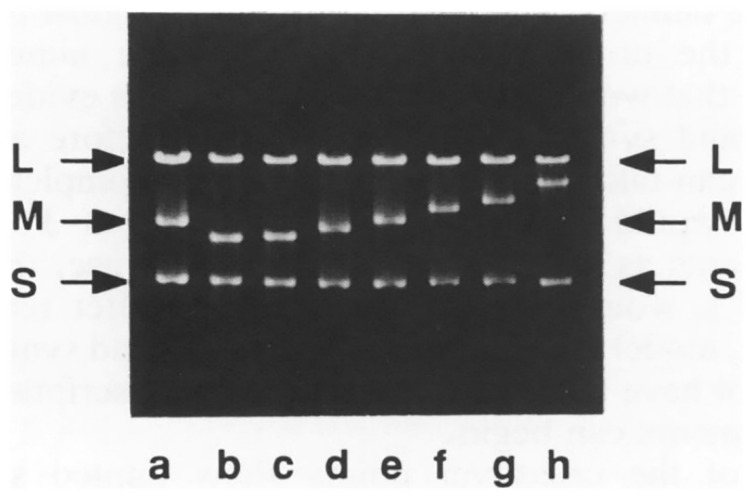
The recombination facilitated by homopolymer base hairpin conformations resulted in an alteration in size of the M dsRNA segment. The agarose electrophoresis gel of dsRNA extracted from recombinant plaques shows a whole variety of sizes of recombinant M segment. Line a shows the dsRNA extracted from φ6. Lines b–h are from recombinant plaques. Upper case denotes dsRNA segments. Reproduced from Onodera et al. (1993).

**Figure 5 viruses-14-02589-f005:**
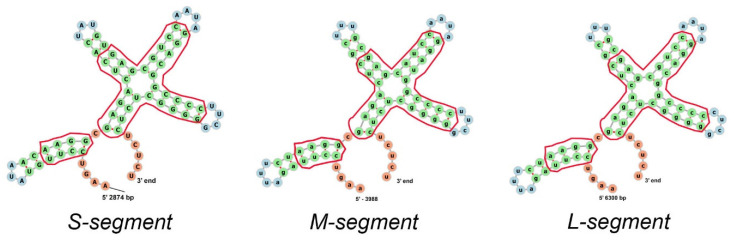
Predicted ssRNA secondary structure of terminal 75 bases at 3′ end for three genomic segments. The region of similarity covers the last 75 nucleotides at the 3′ end and starts with AAGU for all three strands. The structurally identical regions are boxed in red outlines. Note that even nonidentical nucleotides kept a similar secondary structure. The calculation was performed using the *forna* online RNA secondary structure visualization tool (http://rna.tbi.univie.ac.at/forna/, accessed on 17 August 2022). The sequence identity does not significantly affect recombination.

**Figure 6 viruses-14-02589-f006:**
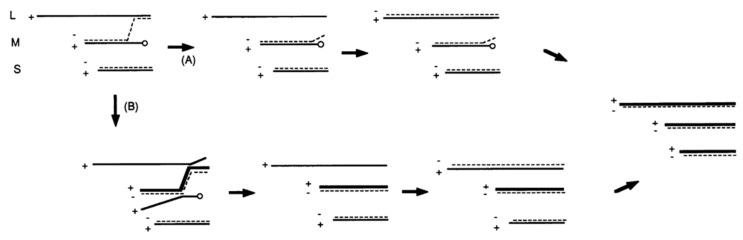
Copy choice model for repair of the M segment. (**A**) The new 3′ end is derived from the l segment when it is used as a template, and the continuation of synthesis shifts to the m plus strand. L synthesis then commences once again and continues to completion. (**B**) Pathway B is more complex in that transcription of the new plus strands displaces the chimeric minus strand from its original m template. The original template would then reinitiate minus-strand synthesis, which proceeds normally. Reproduced from Onodera et al. (1993).

**Figure 7 viruses-14-02589-f007:**
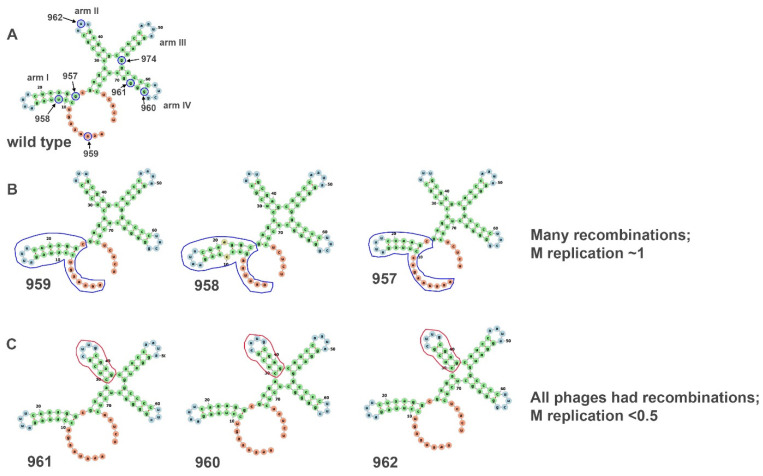
Alteration of the secondary structure of the 3′ end of the M segment. Introduced deletions of the 3′ end caused changes in the predicted secondary structure. The deleted nucleotides are encircled in blue in (**A**). (**B**) The blue outlined arm shows the difference in structure between the wildtype M segment and mutated ones. Changes in the blue outlined arms caused many recombinants in the M segments, but the replication rate of the M segment was nearly the same as of wildtype M. (**C**) The structure of the red outlined hairpin on these three mutants was different from the second arm in wildtype M. These mutations caused a significant reduction in M replication rate (<5), and all the produced phages were recombinant. Based on from Mindich et al. (1994). The secondary structure was calculated using the *forna* online RNA secondary structure visualization tool (http://rna.tbi.univie.ac.at/forna/, accessed on 17 August 2022).

**Figure 8 viruses-14-02589-f008:**
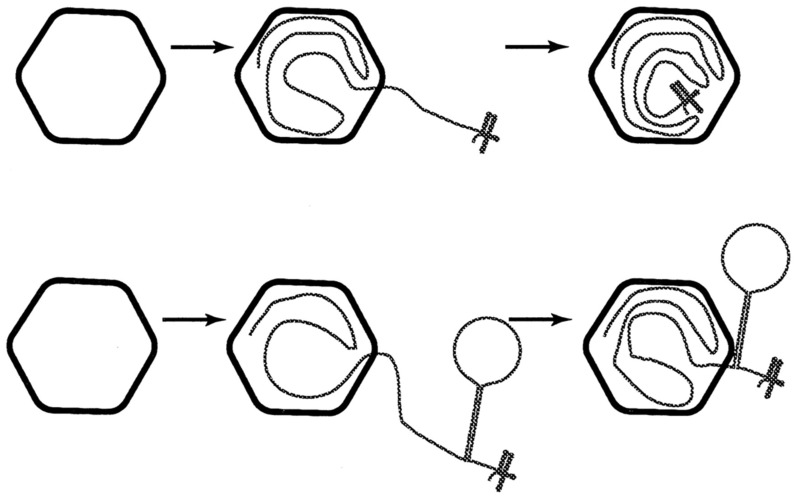
Model showing that the packaging process of ssRNA segment with strong hairpin near the 3′ end can block the process. The exposed hairpin structure becomes vulnerable to RNase I digestion. Reproduced from Qiao et al. (1995).

**Figure 9 viruses-14-02589-f009:**
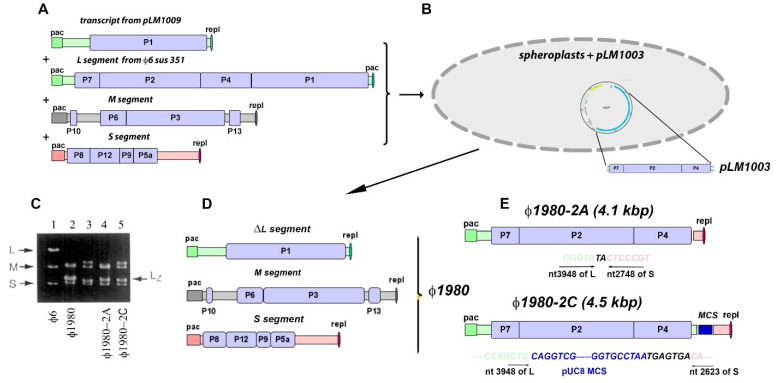
Recombinant phages can acquire the missing genes from the donor plasmid expressed in Pseudomonas host cells. (**A**) Transcripts of partial segment of L (pLM1009), isolated l segment from mutant carrying amber mutation from φ6 sus351, and complete ssRNA from m and s segments of φ6 are in vitro packaged in PC and coated with P8, transducing the spheroplasts of HB10Y with plasmid pLM1003, coding the missing 2, 4, and 7 genes and MCS of pUC8 (**B**). The phage φ1980 carries at least two L segments and M and S segments (**C**,**D**). On the dsRNA agarose gel picture reproduced from Onodera et al. (1995) (**C**), the pattern of φ1980 dsRNA shows a higher intensity of the L band compared to any of the dsRNA bands from the control φ6 dsRNA pattern. If the φ1980 was plated on HB10Y host without donor plasmid pLM1003, the extremely rare lytic plaques were all recombinants. (**E**) The phages packed an extra segment bearing missing 2, 4, and 7 genes and the replication signal from either the M or the S segment. The recombinants had minimal homology (the homology regions are shown as bold black symbols, whereas the recombined RNA origins are shown as light green for the L segment, light pink for S, and blue for pUC8 MCS). Based on Onodera et al. (1995).

**Figure 10 viruses-14-02589-f010:**
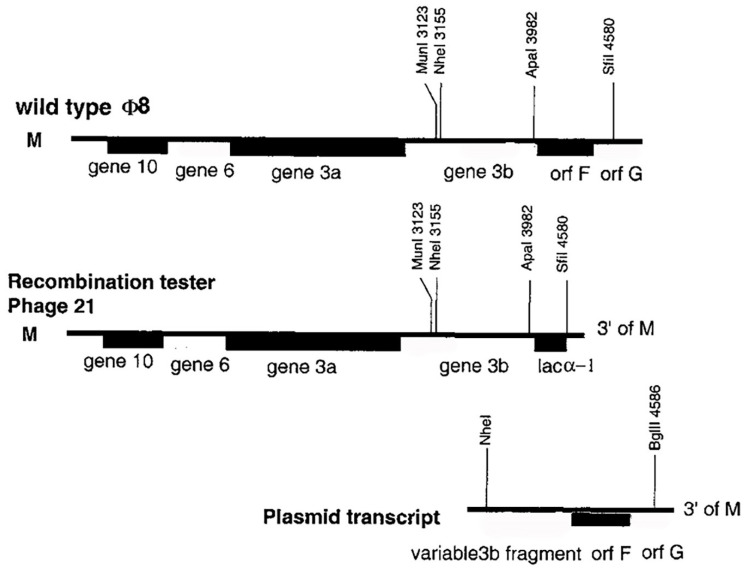
The cDNA copy of φ8 M and the tester phage 21 with lacα in place of genes F and G. The complemented plasmid transcript was of variable size compared to the 3b gene. Reproduced from Onodera et al. (2001).

**Figure 11 viruses-14-02589-f011:**
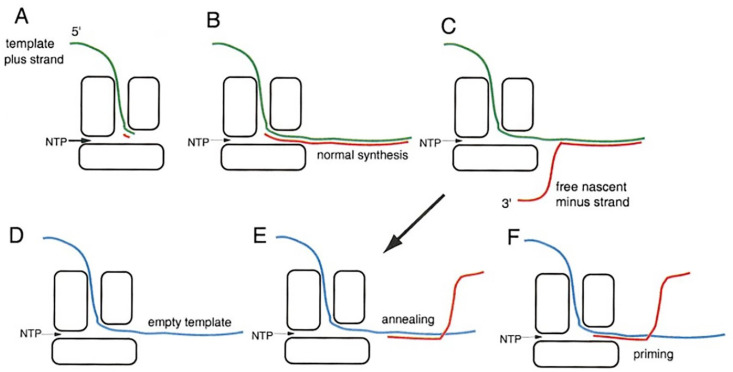
Proposed model for the process of template switching recombination in φ8. Normally, the 3′ end of the template plus strand (green) enters the polymerase and is arrested near the site of nascent minus-strand chain formation (**A**); as the minus strand (red) is synthesized the dsRNA leaves the polymerase through the exit pore (**B**). The 3′ end of the nascent chain can be displaced from the template (**C**). It can reanneal to the template, or it can anneal to an empty template (blue) (**E**). A plus-strand RNA that does not have a proper 3′ end can enter the polymerase, but the template is not arrested at the catalytic site (**D**). Instead, it passes through the polymerase and can scan backward and forward. When a nascent minus strand anneals to the empty template and the template moves back into the polymerase, the nascent chain can act as a primer to start minus-strand synthesis on the empty template (**F**). The result is a recombinant minus strand. Reproduced from Onodera et al. (2001).

**Figure 12 viruses-14-02589-f012:**
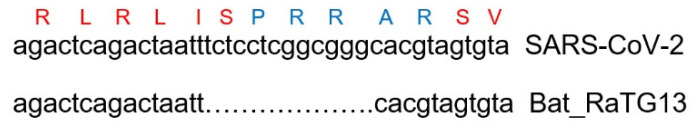
Furin site insertion “PRRAR” into SARS-CoV-2 from Bat_RaTG13. Based on Gallaher (2020).

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
