# Peer review of "Heterologous RNA Recombination in the Cystoviruses φ6 and φ8: A Mechanism of Viral Variation and Genome Repair"

_viruses, 2022, doi:10.3390/v14112589_

Round 1
Reviewer 1 Report
Recombination in RNA viruses is a powerful means whereby these viruses can both compensate for their low fidelity replication and evolve by obtaining new coding sequences from other viruses. The specific molecular mechanisms at play are also a somewhat understudied field when it comes to RNA viruses infecting eucaryotic hosts, where the concept is alive and well. In particular, recombination likely accounts for some of the benefits associated with simultaneous co-infection of multiple virions from the same virus to cancel out deleterious mutations in individual genomes. However, most studies detect the outcome of recombination via selection criteria and seek to identify genome regions or local sequences that may be prone to recombination, but few studies address the exact molecular mechanisms underpinning these effects. This is in large part due to the difficulties of generating a genetically rigorous replication system in eucaryotic hosts, something that is easier to do in procaryotic cells infected by bacteriophage. This historical review nicely summarizes research from Leonard Mindich’s group addressing replication and recombination of the positive-strand RNA cystoviruses in bacteria, an extensive body of work that has provided a number of key insights into RNA virus replication that has directly implications for understanding the biology of RNA viruses with eucaryotic hosts.
While the overall review is of high quality and provides a nice step-by-step (paper-by-paper) summary of prior results, it does seem to start out in a somewhat dated manner with the seven of the first eight references cited being about twenty years old; it would seem that the introductory paragraph could provide some more recent references, before the authors clearly set the stage that the rest of this review is a historical perspective presented in chronological order.
The molecular mechanistic model for recombination presented in Section 11 nicely incorporates aspects of the phi6 polymerase structure and accounts for the path of the template through defined channel structures that likely account for the high processivity of the replication complex. However, this model is fundamentally different from what is proposed in Section 10.3a where the implication is that a polymerase-primer complex seeks out a new template. In the context of this review, these two mechanism models have come out in different papers at different times, but I do think it would appropriate for the authors of this review to discuss the contradictions of the two models in at least a little detail.
The final section of CoV recombination at SARs-CoV-2 emergence is only tangentially related to the cystovirus focus of the rest of the article. It reads like it was simply tacked on to bring in some current relevance, but the discussion is a bit abbreviated and the ideas are not that deeply developed.
Minor issues through the document are as follows:
• Some work on the layout should be done to ensure figure legends do not cross page breaks.
- Typo in line 353
- line 484 – may be nice to add 5’ and 3’ end labels to Figure 8
- line 523, reference 58 should be replaced with 57
- line 576/577 – typos associating with strikethrough text and editing.
- Section 10 as a whole should be proofread to eliminate awkward phrasing and some typos.
- line 675 – consider adding “3’ end”, as in “…the leading edge 3’ end” for parallel construction with the discussion of the 5’ end.
- line 684 – add comma after PC
- lines 784-790 are fairly redundant with the Figure 11 legend.
- line 899 – has it been experimentally shown that the proposed stem-loops in the CoV genome are indeed “capable” of blocking the polymerase complex?
Reviewer 2 Report
Review by Gottlieb and Alimova gives an in depth review on cystovirus RNA recombination in a historical storyline. It is a valuable review not only for phage biologists but for RNA virologists in general. As the authors state at the end of the review it provides a detailed guide for studying intragenomic RNA recombination in viral pathogens. The review excellences in reviewing the studies on replication of dsRNA phages and the possibilities the system offers for genetic studies.
However, I would suggest the authors to still go through the text and make it more flowing. Even though the listing of e.g. the major conclusions of studies is ok, in some parts of the review the text is presented in a list-like manner. In addition, there are several unpolished details in the text:
The presentation of the segment names is not uniform, sometimes the atuhors use capital letters and some parts in lower case.
Most of the figure legends include a reference that does not follow the style of the rest of the manuscript
More specifically:
Line 137: Please rephrase this statement as the latter part of the paragraph conflicts with it (not all cystoviruses infect P. syringae).
inter alia line 321 -> correction has not been done
Line 328: limited number of plaques phage stocks? word missing?
Line 367 and following: Rephrase the presentation of the pathways (not using A. The new…B. Pathway B….
Under section 9 the paragraph division is in the middle of the text covering radiactive counts -> please correct
Line 641: Something wrong
Line 771: Use reference to the figure not “:”
Author Response
please see attachment below
